# Loss of *Resf1* reduces the efficiency of embryonic stem cell self-renewal and germline entry

Matúš Vojtek[1,2] , Ian Chambers[1,2]

Retroelement silencing factor 1 (RESF1) interacts with the key regulators of mouse embryonic stem cells (ESCs) OCT4 and NANOG, and its absence results in sterility of mice. However, the function of RESF1 in ESCs and germline specification is poorly understood. In this study, we used *Resf1* knockout cell lines to determine the requirements of RESF1 for ESC self-renewal and for in vitro specification of ESCs into primordial germ cell-like cells (PGCLCs). We found that deletion of *Resf1* in ESCs cultured in serum and LIF reduces self-renewal potential, whereas episomal expression of RESF1 has a modest positive effect on ESC self-renewal. In addition, RESF1 is not required for the capacity of NANOG and its downstream target ESRRB to drive self-renewal in the absence of LIF. However, *Resf1* deletion reduces the efficiency of PGCLC differentiation in vitro. These results identify Resf1 as a novel player in the regulation of pluripotent stem cells and germ cell specification.

## Introduction

Pluripotency is a feature of early embryonic epiblast and derivative cell lines (Martello & Smith, 2014). Pluripotent cells exist in naïve or primed states (Nichols & Smith, 2009), or an intermediate formative state (Kinoshita & Smith, 2018), from which cells directly differentiate into the germline. Of these pluripotency states, naïve embryonic stem cells (ESCs) are the best characterised (Martello & Smith, 2014). ESC identity is controlled by a gene regulatory network (GRN) centred around Oct4 (*Pou5f1*), Sox2, and Nanog. Whereas Oct4 and Sox2 are uniformly expressed in all pluripotent states, Nanog expression is reduced at the peri-implantation formative state (Chambers et al, 2003; Osorno et al, 2012).

Both the germline and the naïve epiblast show dependencies on NANOG: constitutive *Nanog* deletion prevents specification of the naïve epiblast (Mitsui et al, 2003; Silva et al, 2009), whereas germline-specific *Nanog* deletion reduces the number of primordial germ cells (PGCs) in mid-gestation mouse embryos (Chambers

et al, 2007; Zhang et al, 2018a). On the other hand, Nanog overexpression sustains ESC self-renewal in the absence of the otherwise requisite leukemia inhibitory factor (LIF) (Chambers et al, 2003; Mitsui et al, 2003). Indeed, the level of NANOG expression determines the efficiency of ESC self-renewal, with *Nanog*$^{-/-}$ ESCs having a reduced but residual self-renewal efficiency and *Nanog*$^{+/-}$ ESCs having an intermediate self-renewal efficiency (Chambers et al, 2007). Nanog overexpression can also induce specification of germline competent epiblast-like cells (EpiLCs) into PGC-like cells (PGCLCs) in vitro without the otherwise requisite cytokines (Murakami et al, 2016). Similarities between the GRN of ESCs and PGCs are also highlighted by the capacity of the NANOG target gene, *Esrrb*, to maintain LIF-independent self-renewal in *Nanog*$^{-/-}$ ESCs and to restore wild-type PGC numbers in mouse embryos where *Nanog* was specifically deleted from the germline (Zhang et al, 2018a).

Nanog interacts with more than 100 proteins in ESCs (Gagliardi et al, 2013). However, the requirements of these interactions for NANOG function and ESC self-renewal are largely unknown. Here we examine the function of the NANOG partner protein, RESF1 (also known as KIAA1551, GET) in ESC self-renewal and germline specification. Our findings show that RESF1 has a modest positive effect on ESC self-renewal and that absence of RESF1 decreases efficiency of germline specification.

## Results

### *Resf1* deletion reduces ESC self-renewal and responsiveness of ESCs to LIF

Retroelement silencing factor 1 (RESF1) is a poorly characterised protein that interacts with the core pluripotency proteins OCT4 and NANOG in ESCs (Gagliardi et al, 2013; van den Berg et al, 2010). To study the function of RESF1 in mouse ESC self-renewal, we generated *Resf1*$^{-/-}$ ESCs using CRISPR/Cas9 (Fig 1A). We used two sets of four gRNAs positioned upstream of the transcription start site and downstream of the polyadenylation signal to delete the entire *Resf1* gene (Fig 1A). Wild-type E14Tg2a ESCs were transfected with eSpCas9

[1]Centre for Regenerative Medicine, Institute for Regeneration and Repair, University of Edinburgh, Edinburgh, Scotland [2]Institute for Stem Cell Research, School of Biological Sciences, University of Edinburgh, Edinburgh, Scotland

Correspondence: ichambers@ed.ac.uk

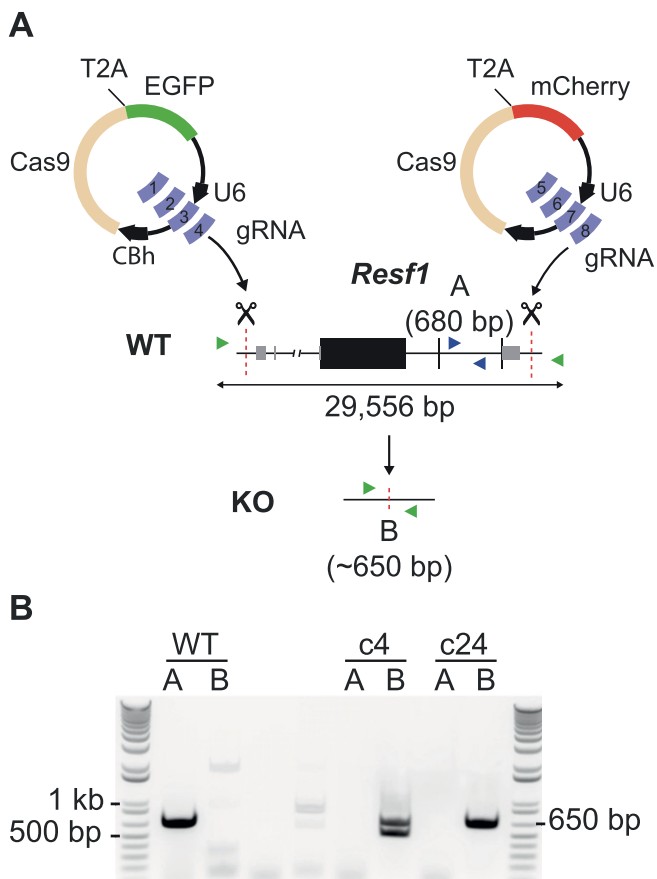

**Figure 1. Deletion of *Resf1* in embryonic stem cells (ESCs).**
**(A)** Scheme of the deletion strategy used at the *Resf1* locus. The line diagram shows *Resf1*, with lines representing introns and non-transcribed regions, thick black boxes represent coding regions of exons and thin grey boxes represent UTRs. *Resf1* was deleted by targeting Cas9 via sets of gRNAs complementary to sites (red dotted lines) lying upstream of the transcription start site and downstream of the polyadenylation signal of the *Resf1* gene. **(A, B)** Deletion of *Resf1* was assessed by PCR using primer pairs nested within intron IV (A, blue triangles) or flanking the targeted sites (B, green triangles). To delete *Resf1*, wild-type (WT) ESCs were transfected with four CBh-eSpCas9-T2a-EGFP and four CBh-eSpCas9-T2a-mCherry plasmids carrying eight distinct gRNAs.
**(B)** Electrophoresis of PCRs from WT ESCs and putative *Resf1⁻/⁻* ESC clones (c4 and c24) using primers A or B.

plasmids encoding individual gRNAs, Cas9, and either GFP or mCherry, depending on whether the gRNA target site was 5′ or 3′ to the *Resf1* gene (Fig 1A). Single cells transiently expressing both GFP and mCherry were isolated and assessed for deletion of *Resf1* by PCR using two primer pairs (Fig 1A). Primer pair A amplifies a 680 bp sequence from the wild-type *Resf1* intron IV (Fig 1A). If *Resf1* is deleted, primer pair A no longer amplifies a product. Primer pair B spans the entire *Resf1* locus (Fig 1A), and a large distance prevents PCR amplification from wild-type cells under the reaction conditions used. However, when *Resf1* is deleted, these primers come into proximity to yield a product of ~650 bp. Thirty clonal cell lines were expanded and analysed. Of these, two clones (c4 and c24) showed patterns of PCR amplification with primers A and B suggesting that they had deleted both copies of the *Resf1* gene (Fig 1B). For c4, the presence of two bands of differing sizes suggested that

each *Resf1* allele had been deleted using different gRNA pairs. Sequencing of the PCR products confirmed this. Sequencing also confirmed that each of the *Resf1* alleles in c24 had undergone distinct deletion events (Fig S1).

The capacity of *Resf1⁻/⁻* ESC clones 4 and 24 to self-renew in the presence or absence of LIF was examined after plating at clonal density. After 6 d in the presence of saturating levels of LIF, *Resf1⁻/⁻* ESCs formed colonies with a similar morphology to the parental wild-type E14Tg2a ESCs (Fig 2A). The proportion of colonies expressing AP was also similar between the examined cell lines. In the absence of LIF, wild-type cells do not produce any uniformly undifferentiated AP+ colonies but do yield a proportion of colonies containing differentiated and undifferentiated AP+ cells. The number of these mixed colonies was significantly reduced in both *Resf1⁻/⁻* clones (Wilcoxon rank-sum test, q < 0.05; Fig 2A and B). This suggests that *Resf1* deletion has a negative effect on ESC self-renewal. To investigate this further, the colony-forming assay was repeated at decreasing concentrations of LIF. *Resf1⁻/⁻* cell lines formed fewer AP+ colonies at all LIF concentrations, with the largest differences observed at, or below 3 U/ml LIF (Wilcoxon rank-sum test, q < 0.05) (Figs 2C and S2A). This suggests that RESF1 sensitises the ESC response to low levels of LIF signalling.

### Deletion of *Resf1* decreases expression of LIFR

To investigate the possible basis for the differential sensitivity to LIF, we analysed published RNA-seq data generated from *Resf1⁻/⁻* and wild-type ESCs (Fukuda et al, 2018). Initial analysis suggested that LIF receptor (LIFR) expression may be down-regulated in *Resf1⁻/⁻* cells relative to wild-type ESCs. However, the LIFR gene expresses distinct mRNAs whose principal functional distinction is their capacity to encode transmembrane LIFR or a soluble form of LIFR. Soluble LIFR mRNA lacks the last five exons encoding the membrane-spanning and cytoplasmic domains of transmembrane LIFR mRNA (Chambers et al, 1997), with soluble LIFR acting as an LIF antagonist (Layton et al, 1992; Tomida, 1995). As soluble and transmembrane LIFR mRNAs can be independently regulated (Chambers et al, 1997), we quantified the RNA-seq reads per transcript rather than per gene to differentiate between these isoforms. This suggested that a transmembrane form of LIFR mRNA (ENSMUST00000171588) was down-regulated ~2.5-fold in *Resf1⁻/⁻* ESCs (adjusted *P*-value 4.28 × 10⁻¹², DESeq2) compared with wild-type ESCs (Fig 2D).

To directly assess the relative expression of soluble and transmembrane forms of LIFR mRNA, quantitative PCR of reverse-transcribed RNA isolated from *Resf1⁻/⁻* and wild-type ESCs was performed using primers that discriminate between soluble and transmembrane LIFR mRNAs (Fig 2E). This established that both forms of LIFR mRNA were expressed ~2.5-fold less in *Resf1⁻/⁻* ESCs than in wild-type ESCs (LIFR: q = 0.056 and sLIFR: q = 0.036, two-tailed *t* test) (Fig 2E). Consistent with previous findings in ESCs, transmembrane LIFR mRNA was expressed at far higher levels than soluble LIFR mRNA (Chambers et al, 1997). The co-ordinate changes in expression of soluble and transmembrane *LIFR* mRNAs is consistent with a mechanism in which RESF1 affects ESC self-renewal by acting (directly or indirectly) to stimulate transcription of the *LIFR* gene.

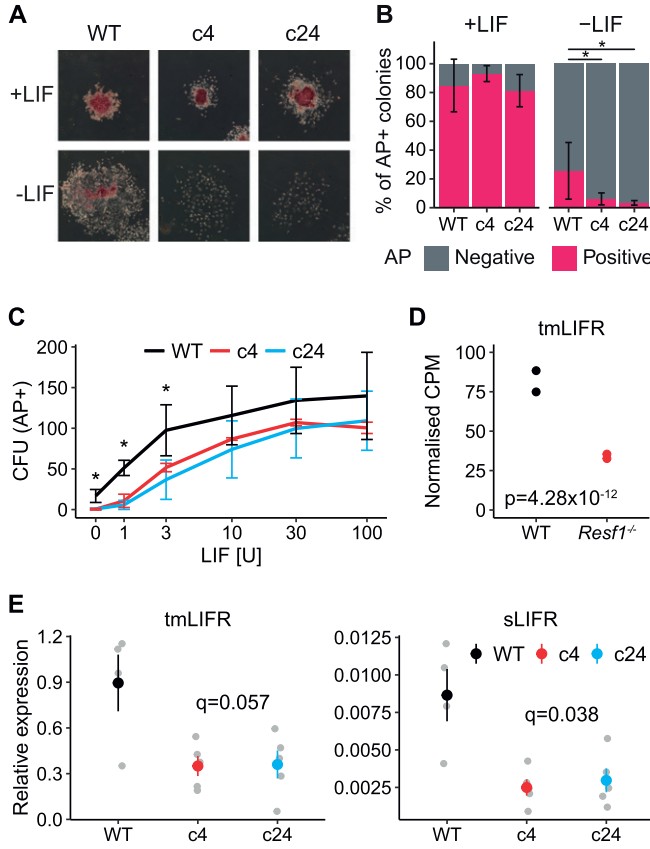

**Figure 2.** Deletion of *Resf1* reduces embryonic stem cell (ESC) self-renewal by decreasing expression of LIFR.
**(A, B, C)** Clonal ESC self-renewal assays. **(A)** Representative images of the colonies formed by the indicated ESCs in the presence or absence of leukemia inhibitory factor (LIF). Colonies were stained for AP 6 d after plating. **(B)** Proportion of AP+ colonies formed by WT, *Resf1⁻/⁻* c4, and *Resf1⁻/⁻* c24 ESCs in the presence or absence of LIF. Bars represent mean ± SD (n = 5; *q < 0.05; Wilcoxon rank-sum test). **(C)** Number of AP+ CFU generated by WT or *Resf1⁻/⁻* ESCs at different LIF concentrations (U/ml); mean ± SD (n = 4). **(D)** CPM for ENSMUST00000171588 transmembrane LIF receptor (tmLIFR) transcript in wild-type and *Resf1⁻/⁻* ESCs; n = 2 RNA-seq data from Fukuda et al (2018). Adjusted *P*-value calculated by DESeq is shown, **(E)** Quantification of tmLIFR and soluble LIFR (sLIFR) transcript levels in wild-type, *Resf1⁻/⁻* c4, and *Resf1⁻/⁻* c24 ESCs by quantitative PCR on reverse-transcribed RNA. Grey points show individual data points. Coloured point ranges represent mean ± SE; n = 4 for wild-type cells, n = 5 for *Resf1⁻/⁻* cells. FDR corrected *P*-values are shown (two-tailed *t* test).

## Deletion of *Resf1* decreases transcription factor expression in ESCs cultured in serum

To further assess the reduced self-renewal efficiency of *Resf1⁻/⁻* ESCs, the expression of key pluripotency transcription factors was examined in *Resf1⁻/⁻* cells (Fig 3A). In both *Resf1⁻/⁻* ESC clones, mRNA levels of *Nanog*, *Esrrb*, *Klf4*, and *Pou5f1* were reduced compared with the wild-type ESCs, with *Esrrb* mRNA levels reduced by ~4-fold (*t* test, q < 0.05; Fig 3A). As previously reported, deletion of *Resf1* has a large impact on transcription in ESCs (Fukuda et al, 2018). Consistent with our RT-qPCR experiments, *Esrrb* and *Klf4* were significantly down-regulated in *Resf1⁻/⁻* ESCs (*P*.adj. < 0.05, fold change > 1.5; Fig 3B). However, *Pou5f1* and *Nanog* were not differentially expressed (*P*.adj. < 0.05, fold change > 1.5; Fig 3B). This is

in line with the lower level of down-regulation of *Pou5f1* and *Nanog*, compared with *Esrrb* in our RT-qPCR data.

ESCs cultured in serum/LIF medium are heterogeneous for NANOG and ESRRB expression (Chambers et al, 2007; Festuccia et al, 2012). This heterogeneity can be eliminated in culture media containing two small inhibitors (2i) blocking FGF signalling and GSK3β (Silva & Smith, 2008; Ying et al, 2008). After switching to 2i/LIF culture, mRNA levels of *Esrrb*, *Nanog*, *Pouf51*, or *Rex1* became equivalent in wild-type and *Resf1⁻/⁻* ESCs (Fig 3C). This suggests that the reductions observed in serum/LIF may result from the cells initiating differentiation in serum/LIF culture.

### Episomal expression of RESF1 has a modest positive effect on ESC self-renewal

As *Resf1* deletion reduced ESC self-renewal efficiency and decreased expression of pluripotency transcription factors, we hypothesised that enforced RESF1 expression may increase ESC self-renewal. To test this, Flag-RESF1 was expressed from an episome in *Resf1⁺/⁺* E14/T ESCs (Chambers et al, 2003) (Fig 4A). Transfected cells were cultured in selection medium at clonal density in the presence or absence of LIF for 8 d (Fig 4A), stained for AP activity and quantified (Fig 4B and C). In the absence of LIF, NANOG conferred LIF-independent ESC self-renewal (Chambers et al, 2003), but RESF1 did not (Fig 4C). In contrast, in the presence of LIF, ESCs expressing episomal Flag-RESF1 formed significantly more AP+ colonies than ESCs transfected with an empty vector (Wilcoxon rank-sum test, q < 0.01), although fewer than that obtained after Nanog transfection (Fig 4C).

### *Resf1* is not required for NANOG or ESRRB function in ESC self-renewal

The capacity of ESCs to self-renew is sensitive to NANOG levels (Chambers et al, 2003, 2007; Mitsui et al, 2003). As NANOG interacts with RESF1 (Gagliardi et al, 2013), we investigated the importance of RESF1 for NANOG function in ESC self-renewal. First, we validated the interaction between NANOG and RESF1 by co-immunoprecipitation. Episomally expressed Flag-RESF1 was immunoprecipitated from nuclear extracts of ESCs transfected with plasmids encoding Flag-RESF1 and HA-NANOG. When both Flag-RESF1 and HA-NANOG were co-expressed, Flag immunoprecipitation co-purified HA-NANOG (Fig 5A). In contrast, Flag antibody did not purify HA-NANOG from the control sample lacking Flag-RESF1 (Fig 5A). This confirms that RESF1 and NANOG interact in ESCs.

To examine the importance of RESF1 for NANOG function, we assessed whether *Resf1* was required for NANOG, or its downstream target ESRRB to confer LIF-independent self-renewal. Constitutive transgenes expressing Flag-NANOG-IRES-Puro, Flag-ESRRB-IRES-Puro, or dsRed-IRES-Puro transgenes were stably integrated into wild-type and *Resf1⁻/⁻* ESCs (Fig 5B). After 12-d selection, cell populations transfected with Nanog or Esrrb transgenes expressed Flag-NANOG or Flag-ESRBB (Fig 5C). The self-renewal capacity of these cell lines was next assessed by quantification of colony forming assays. In the presence of LIF, both wild-type and *Resf1⁻/⁻* cells formed similar numbers of AP-positive colonies (Figs 5D and S2B). In the absence of LIF, expression of Flag-NANOG or Flag-ESRRB in wild-type ESCs supported clonal ESC self-renewal (Fig 5D).

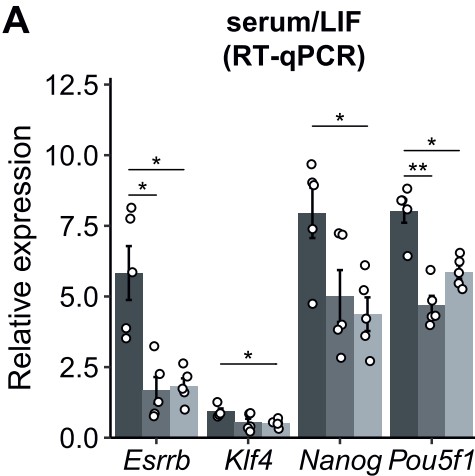

A **serum/LIF (RT-qPCR)**

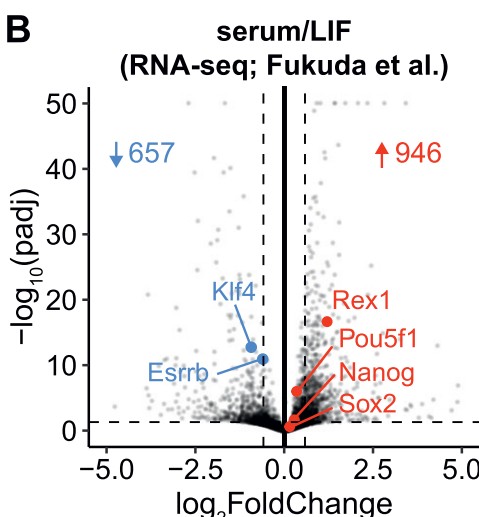

B **serum/LIF (RNA-seq; Fukuda et al.)**

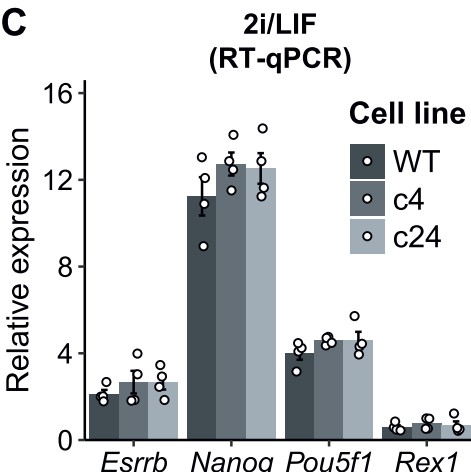

C **2i/LIF (RT-qPCR)**

Although overexpression of Flag-NANOG or Flag-ESRRB in *Resf1*⁻/⁻ cells appeared to increase AP-positive colony formation relative to wild-type ESCs (Fig 5D), this could be due to a higher expression of NANOG and ESRRB transgenes in *Resf1*⁻/⁻ cells (Fig 5C). These results indicate that *Resf1* is not required for ESRRB and NANOG to sustain LIF-independent self-renewal.

### Epitope tagging of endogenous *Resf1*

As there are no available antibodies for mouse RESF1, we generated ESC lines carrying an epitope tagged endogenous *Resf1* gene to facilitate the study of its molecular properties. To do this, we transfected E14Tg2a ESCs with a modification construct and a plasmid carrying a single gRNA and Cas9. The modification construct extended the *Resf1* open reading frame to include three V5 epitope tags, followed by an internal ribosomal entry site (IRES)-GFP cassette and a single loxP site (Fig 6A).

After 2 d, single cells expressing high levels of GFP were isolated, expanded, and genotyped. We used a primer pair flanking the insertion site that produces a 738 bp band from the wild-type allele and that increases in size to 2,362 bp upon insertion of the tagged modification (Fig 6A). Three ESC lines with an extended *Resf1* allele and lacking a wild-type allele were identified (Fig 6B). Immunofluorescence confirmed expression of the v5 epitope in all three cell lines (Fig 6C). RESF1-v5 localized to the nucleus (Fig 6C) consistent with the previous observations using overexpression of RESF1 (Fukuda et al, 2018). Like NANOG, RESF1-v5 is expressed heterogeneously in ESCs cultured in serum/LIF. However, RESF1 is present in a larger subset of ESCs than NANOG, with several NANOG negative cells expressing RESF1-v5 (Fig 6C, white arrows). This is similar to findings for NANOG-TET2 co-expression pattern (Pantier et al, 2019). As *Nanog*-null ESCs are pluripotent (Chambers et al, 2007) this RESF1-positive, NANOG-negative population is predicted to contain pluripotent cells.

### Deletion of *Resf1* decreases efficiency of PGCLC specification

NANOG is required to provide wild-type numbers of PGCs in vivo (Zhang et al, 2018a). In addition, enforced expression of NANOG in epiblast-like cells (EpiLCs) enables cytokine independent differentiation of PGC-like cells (PGCLCs) in vitro (Murakami et al, 2016; Zhang et al, 2018a). RESF1 is reported to be required for fertility in mice, although the mechanism responsible is unknown (Dickinson et al, 2016). Therefore, we investigated the function of RESF1 in early germline specification.

First, we examined *Resf1* mRNA levels during germline specification in vitro. We quantified *Resf1* mRNA levels in wild-type E14Tg2a naïve ESC cultures (serum/LIF and 2i/LIF medium), formative EpiLCs, primed epiblast stem cells (EpiSCs), and PGCLCs. *Resf1* mRNA levels in ESCs, EpiLCs, and EpiSCs were similar, although

**Figure 3. Effect of *Resf1* deletion on the expression of pluripotency markers in naïve pluripotent stem cells.**
**(A)** mRNA levels of the indicated transcripts were determined in wild-type (WT) and *Resf1*⁻/⁻ embryonic stem cells (c4 and c24) cultured in serum/leukemia inhibitory factor (LIF) medium (n = 5). **(B)** Volcano plot comparing transcriptomes of WT and *Resf1*⁻/⁻ embryonic stem cells (Fukuda et al, 2018).

Dashed lines represent significance thresholds. Number of significantly down-regulated (blue) and up-regulated (red) genes are shown. Selected pluripotency transcription factors are highlighted. **(C)** As (A) but cells were maintained in 2i/LIF medium. Bars and whiskers represent mean ± SEM (n = 4). Scatter plots represent individual data points. *q < 0.05 (two-tailed *t* test).

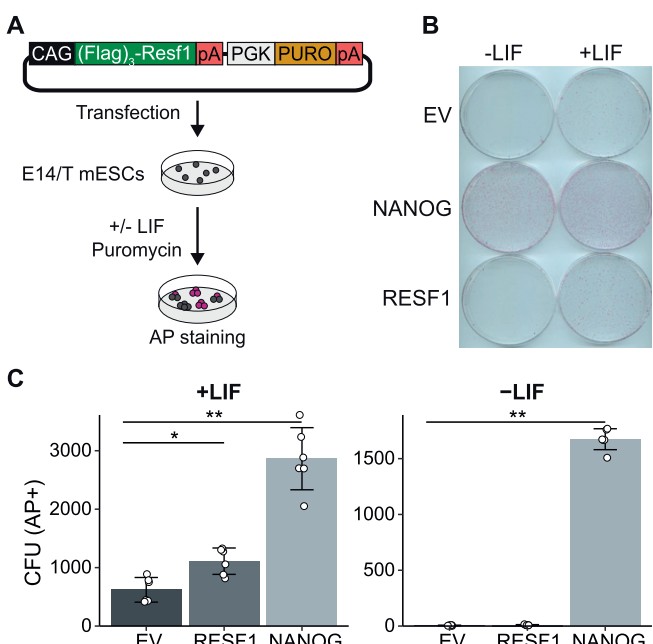

**Figure 4. Effect of episomal expression of Resf1 on embryonic stem cell (ESC) self-renewal.**
**(A)** Strategy to assess the effect of episomal expression of RESF1 on ESC self-renewal. E14/T ESCs were transfected with the plasmid shown and cultured in the presence of puromycin in medium with or without leukemia inhibitory factor (LIF). **(B)** After 8 d, colonies were stained for AP. Empty vector (EV) and a plasmid encoding Nanog in place of Resf1 provided controls. **(C)** Quantification of colony numbers from (B). Bars represent mean ± SD (n = 6) and scatter plots individual data points; **q < 0.01, *q < 0.05 (Wilcoxon rank-sum test).

median *Resf1* expression was higher when ESCs were cultured in serum/LIF medium (Fig S3A). Interestingly, day 4 PGCLCs expressed higher levels of *Resf1* than EpiSCs.

To investigate *Resf1* expression further, we analysed published single cell RNA sequencing datasets from mouse epiblast and PGCs between embryonic days (E) 6.5 and 8.5 (Pijuan-Sala et al, 2019). *Resf1* expression was detected in epiblast cells between E6.5 and E7.75 (Fig S3B). *Resf1* expression in epiblast cells is highest at E6.75 and decreases at later stages (Fig S3C). *Resf1* is also continuously expressed in PGCs between E6.75 and E8.5 (Fig S3B). In agreement with our RT-qPCR results, *Resf1* expression is higher in PGCs than in cells of post-implantation epiblast (Fig S3C). These results suggest that RESF1 might function in early germ cell development.

Therefore, we examined the capacity of *Resf1*[−/−] ESCs to specify PGCLCs in vitro (Fig 7A). Initially, wild-type and *Resf1*[−/−] cells formed EpiLCs with similar morphologies (Fig S4A) and a similar expression of EpiLC markers *Fgf5*, *Otx2*, and *Pou3f1* (Fig S4B), indicating proper transition of *Resf1*[−/−] ESCs into PGC competent EpiLCs. Further aggregation of the wild-type EpiLCs in the presence of PGC-specifying cytokine cocktail for 4 d induced surface expression of CD61 and SSEA1, which jointly mark PGCLCs (Hayashi et al, 2011) (Figs 7B and C and S5). However, the proportion of SSEA1+CD61+ cells in the population was reduced in both clonal R*esf1*[−/−] cell lines (Fig 7B and C). Moreover, expression of mRNAs encoding the key PGC transcription factors *Ap2γ*, *Blimp1*, and *Prdm14* was reduced in *Resf1*[−/−] cells (Fig 7D). This effect was clearest for *Blimp*1, which

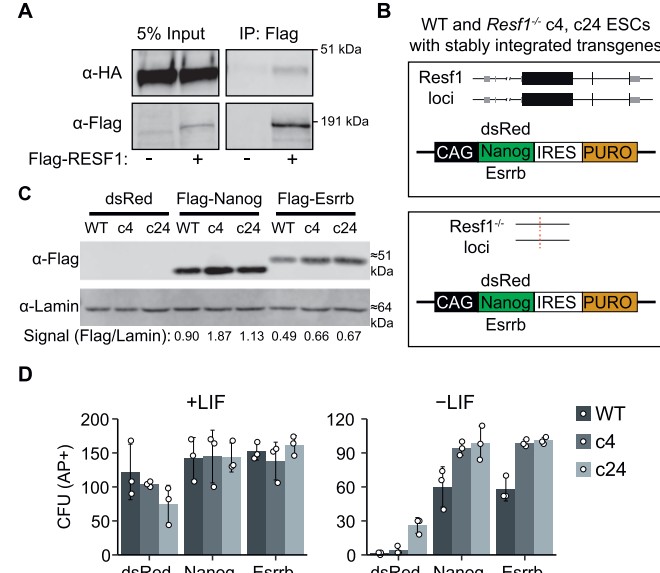

**Figure 5. *Resf1* is dispensable for NANOG and ESRRB function to sustain leukemia inhibitory factor-independent self-renewal.**
**(A)** Co-immunoprecipitation of Flag-RESF1 and HA-NANOG from nuclear extracts of embryonic stem cells (ESCs) episomally expressing HA-NANOG alone (−) or HA-NANOG plus Flag-RESF1 (+). **(B)** Schematic representation of wild-type (WT) or *Resf1*[−/−] ESCs with stably integrated transgenes in which the puromycin resistance gene (PURO) is linked in the same transcript to either dsRed, Flag-Nanog or Flag-Esrrb. Black boxes represent coding exons, grey boxes represent non-coding exons. **(C)** Immunoblot analysis of Flag expression after stable integration of dsRed, Flag-Nanog, or Flag-Esrrb expression cassettes in WT and *Resf1*[−/−] ESCs (c4, c24); anti-LAMIN was used as a loading control. Relative Flag signal over LAMIN control is shown below. **(B, D)** Quantification of alkaline positive (AP+) CFUs formed by cells described in (B) after 8-d culture in the presence or absence of leukemia inhibitory factor. Bars represent mean ± SD (n = 3).

mRNA levels were lowered to ~40% of wild-type expression in both *Resf1*[−/−] clonal cell lines (two-tailed *t* test, q < 0.05; Fig 7D). Together, these results indicate that *Resf1* contributes to efficient PGCLC differentiation in vitro and that the *Resf1* requirement may occur downstream of *Ap2γ* and upstream of *Blimp1* and CD61/SSEA1 expression.

## Discussion

RESF1 is a poorly characterised protein lacking known functional domains. However, the interaction of RESF1 with the pluripotency transcription factors NANOG and OCT4 (van den Berg et al, 2010; Gagliardi et al, 2013) suggests that RESF1 may function in ESC nuclei. Consistent with this, RESF1 has been reported to localise to nuclei when overexpressed in ESCs (Fukuda et al, 2018), a finding we confirm here using ESCs carrying epitope tagged endogenous *Resf1* loci.

A previous study showed that *Resf1*[−/−] ESCs could be maintained in serum/LIF culture (Fukuda et al, 2018), suggesting that ESC self-renewal continues after *Resf1* deletion. We confirm this here using clonal ESC self-renewal assays performed at saturating LIF concentrations. We have extended these observations by showing that *Resf1*[−/−] ESCs cultured at low LIF concentrations show reduced self-renewal compared

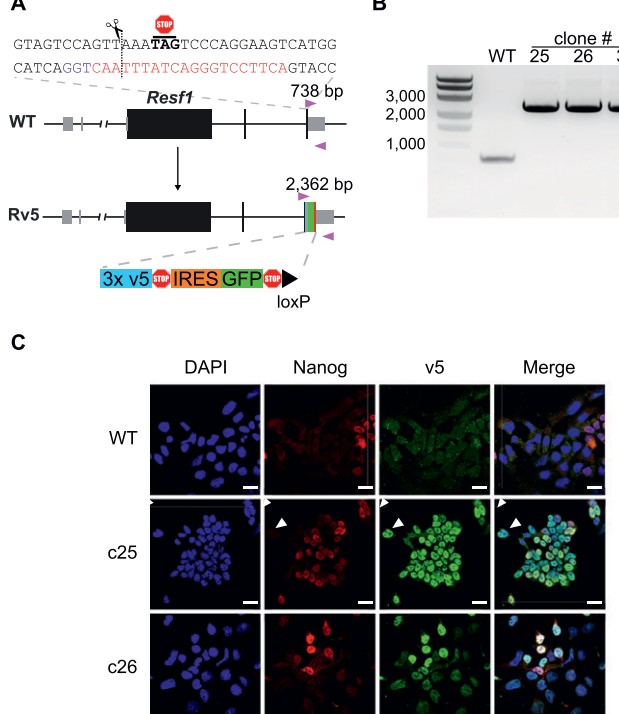

**Figure 6. Epitope tagging of the endogenous *Resf1* gene.**
**(A)** Cartoon representation of *Resf1* in wild-type (WT) and Resf1-tagged (Rv5) cells. Lines represent introns and non-transcribed regions; thick boxes represent coding exons; thin grey boxes represent non-coding exons. *Resf1* was tagged by inserting 3× v5-tag epitopes, stop codon, internal ribosome entry site, GFP, stop codon, and a loxP site in front of the *Resf1* stop codon using CRISPR/Cas9. The complementary sequence of the gRNA (red) with the PAM sequence (blue) is shown. The expected cleavage site is indicated by a dotted line close to the *Resf1* stop codon. Genotyping primers (pink triangles) and expected PCR product sizes are shown. **(B)** Genotyping of WT and Rv5 clones using primers flanking the insertion site. **(C)** Immunostaining of WT and Rv5 clones for RESF1-v5 and NANOG. White arrows indicate cells expressing v5 but no NANOG. White scale bars represent 25 μm.

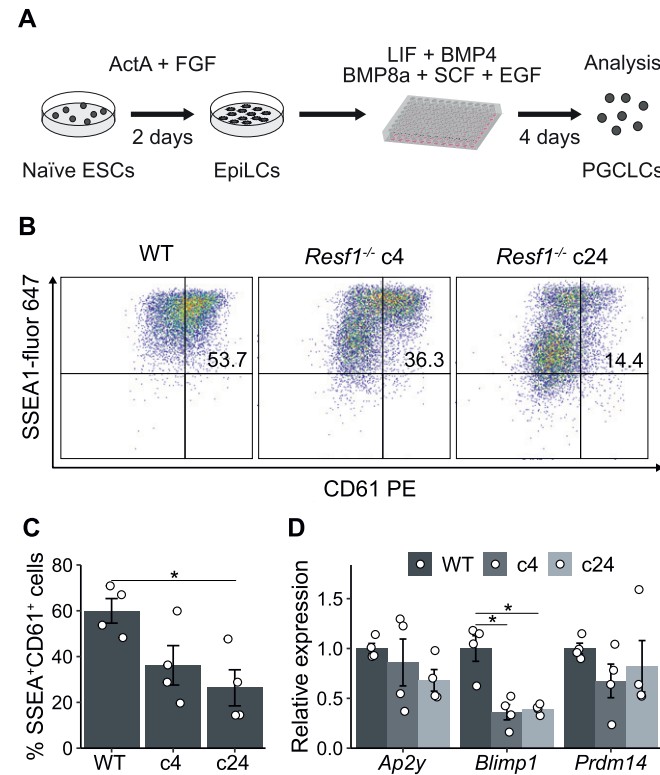

**Figure 7. Effect of *Resf1* deletion on PGCLC differentiation.**
**(A)** Scheme of differentiation of naïve embryonic stem cells into primordial germ cell-like cells (PGCLCs). Embryonic stem cells (ESCs) are treated with Activin A and Fgf2 for 2 d to form EpiLCs. EpiLCs are then aggregated in the presence of the indicated cytokines. **(B)** Representative scatter plots of SSEA1 and CD61 expression measured by flow cytometry after 4 d of PGCLC differentiation using the indicated cell lines. Numbers represent percentage of CD61⁺SSEA1⁺ population. **(B, C)** Quantification of CD61⁺SSEA1⁺ cell populations shown in (B) (n = 4). **(D)** Relative expression of the indicated primordial germ cell markers in WT or *Resf1⁻/⁻* cells after 4 d of PGCLC differentiation (n = 4). Bars represent mean ± SEM. Scatter plots represent individual data points. *q < 0.05 (two-tailed *t* test).

with wild-type ESCs. Moreover, RESF1 overexpression increased colony forming capacity of ESCs cultured in serum/LIF. Interestingly, the levels of the key pluripotency mRNAs *Nanog*, *Esrrb*, *Klf4*, and *Pou5f1* were reduced in *Resf1⁻/⁻* ESCs cultured in serum/LIF, an effect that was reversed by culture of *Resf1⁻/⁻* ESCs in LIF media supplemented with MEK and GSK3β inhibitors (2i). This suggests that RESF1 causes ESCs to become either more responsive to LIF or Wnt or less sensitive to MEK. This is consistent with our gene expression analysis, which indicates that RESF1 augments ESC self-renewal by stimulating LIFR expression. Together, these results suggest that RESF1 has a positive effect on ESC self-renewal.

As the only previously reported function of RESF1 is to silence endogenous retroviruses (ERVs), the different effects of *Resf1* deletion in ESCs cultured in 2i/LIF and serum/LIF could also relate to the different activities of ERVs in these conditions. ESCs express higher levels of IAPEy and ERVK mRNAs when cultured in 2i/LIF

rather than serum/LIF (Hackett et al, 2017). Interestingly, ESCs express lower levels of *Resf1* mRNA when cultured in 2i/LIF rather than serum/LIF, suggestive of a reciprocal relationship. Therefore, RESF1 function may be less critical during 2i/LIF culture. However, further investigation is needed to determine the basis of higher tolerance of ESCs cultured in 2i/LIF to RESF1 depletion and ERV activity.

RESF1 also interacts with the histone methyltransferase SETDB1 (Fukuda et al, 2018). In *Resf1⁻/⁻* ESCs, a decrease in SETDB1 binding and trimethylated histone 3, lysine 9 (H3K9me3) at an integrated murine stem cell virus reporter locus suggests that RESF1 may support chromatin binding of SETDB1 (Fukuda et al, 2018). Similarly to *Resf1* deletion, deletion of SETDB1 or its associated protein TRIM28 leads to down-regulation of the key pluripotency genes *Pou5f1*, *Sox2*, and *Nanog* (Bilodeau et al, 2009; Hu et al, 2009; Yuan et al, 2009; Rowe et al, 2010; Karimi et al, 2011). However, deletion of SETDB1 or TRIM28 also resulted in ESC differentiation (Bilodeau et al, 2009; Hu et al, 2009; Yuan et al, 2009), indicating a more pronounced effect of SETDB1 and TRIM28 on ESC self-renewal than RESF1. Moreover, a recent report indicates that another SETDB1

partner YTHDC1, that also silences retrotransposons in ESCs is critically required for ESC self-renewal (Liu et al, 2021). As the functions of RESF1 and YTHDC1 overlap, this suggests that functions of YTHDC1 not shared by RESF1 are required for ESC self-renewal. Nevertheless, an interaction with SETDB1 may be required for RESF1 to support efficient ESC self-renewal.

A further indication that RESF1 may participate in the pluripotency GRN comes from the binding of RESF1 to NANOG and OCT4 (van den Berg et al, 2010; Gagliardi et al, 2013). To better understand the molecular function of RESF1 in ESC self-renewal, we examined the relationship between RESF1 and NANOG. We validated the interaction of RESF1 and NANOG in ESC nuclei. However, this interaction is not essential for NANOG to sustain LIF-independent self-renewal as $Resf1^{-/-}$ ESCs self-renew in the absence of LIF after NANOG overexpression. Also, $Resf1^{-/-}$ ESCs overexpressing the NANOG downstream target ESRRB also sustained LIF-independent self-renewal. Therefore, RESF1 is not required for NANOG or ESSRB to sustain LIF-independent ESC self-renewal.

NANOG and RESF1 both function in the germline (Chambers et al, 2007; Dickinson et al, 2016; Zhang et al, 2018a). NANOG is required to provide wild-type numbers of PGCs and is able to confer germline specification in the absence of instructive external signals (Murakami et al, 2016), whereas $Resf1$ deletion causes infertility in both male and female mice (Dickinson et al, 2016). RESF1 may function during late germline development (Hudson et al, 2005) as deletion of the RESF1 partner protein SETDB1 prevents ERV silencing in E13.5 PGCs and blocks germline development (Liu et al, 2014). However, $Resf1$ mRNA was also detected in PGCs at E6.75 and at higher levels than in epiblast cells at this stage, suggestive of a possible role in PGC specification. Indeed, $Resf1^{-/-}$ cells differentiate into PGCLCs with a lower efficiency in vitro, as judged by SSEA1 and CD61 expression, and express lower levels of the key PGC transcription factor $Blimp1$. This suggests that RESF1 functions early during PGC specification.

Our results suggest that RESF1 has a modest effect on ESC self-renewal and plays a role in early germline specification. It will be interesting to investigate the contribution of RESF1 to germline development in vivo because $Resf1$-null mice are infertile. Furthermore, assessing RESF1 function during human germline development could have implications for reproductive medicine. Our mouse ESC lines carrying endogenously labelled $Resf1$ alleles may be a valuable molecular tool for further deciphering the function of RESF1 in vivo.

## Materials and Methods

### Cell culture

All cell lines were derived from E14Tg2a ESCs (Hooper et al, 1987) and were routinely cultured on 0.1% gelatin coated plates in serum/LIF medium (Glasgow Minimum Essential Medium [G5154; Sigma-Aldrich], 10% foetal calf serum, 1× L-glutamine [25030-024; Invitrogen], 1× pyruvate solution [11360-039; Invitrogen], 1× MEM non-essential amino acids [11140-036; Invitrogen], 0.1 mM 2-mercaptoethanol [31350010; Gibco], and 100 U/ml LIF [homemade]) at 37°C, 7% $CO_2$.

Cells were passaged every other day using trypsin solution (0.372 mg/ml EDTA [Cat. no. E5134; Sigma-Aldrich], 1% chicken serum [Cat. no. C5405; Sigma-Aldrich], and 0.025% wt/vol trypsin [Cat. no. 15090-046; Invitrogen]).

For 2i/LIF culture, ESCs cultured in serum/LIF medium were adapted to serum-free N2B27 medium supplemented with 3 $\mu$M CHIR99021 (Cat. no. 1677-5; Cambridge Bioscience), 0.4 $\mu$M PD0325901 (Cat. no. 72182; STEMCELL Technologies), and 100 U/ml homemade LIF as described by Hayashi and Saitou (2013) for at least three passages. Cells were passaged on plates pre-treated with 0.01% wt/vol poly-L-ornithine and coated with 10 ng/ml laminin (Cat. no. 354232; BD Biosciences).

ESCs were differentiated into EpiSCs as previously described (Guo et al, 2009). 3 × $10^4$ ESCs cultured in serum/LIF medium were plated on a gelatin-coated six-well plate and cultured for 1 d. Cells were washed twice with PBS and further cultured in EpiSC medium made by supplementing N2B27 medium with 20 ng/ml Fgf2 (Cat. no. 233-FB-025/CF; R&D Systems) and 20 ng/ml Activin A (Cat. no. 120-14E; PeproTech). After 24 h, cells were washed twice using PBS and dissociated by 200 $\mu$l TrypLE Express (Cat. no. 12604013; Gibco) for 2 min at 37°C. Cells were further cultured on plates coated with 7.5 $\mu$g/ml Fibronectin (Cat. no. F1141; Sigma-Aldrich) in EpiSCs medium. EpiSCs were analysed after six passages.

ESCs were differentiated into PGCLCs as previously described (Hayashi & Saitou, 2013; Zhang et al, 2018b). ESCs cultured in 2i/LIF medium were treated with TrypLE Express (Cat, no. 12604013; Gibco) to obtain a single cell suspension. 1 × $10^5$ ESCs were plated on a 3.8-cm$^2$ plate coated with human plasma fibronectin (Cat. no. FC010; Millipore) and cultured in N2B27 medium supplemented with 12 ng/ml Fgf2 (Cat. no. 233-FB-025/CF; R&D Systems), 20 ng/ml Activin A (Cat. no. 120-14E; PeproTech), and 1% KSR (Cat. no. 10828028; Gibco) for 44 h. Cells were treated with TrypLE Express and resuspended in GK15 medium (GMEM [Cat. no. G5154; Sigma-Aldrich], 15% KSR [Cat. no. 10828-028; Invitrogen], 1× nonessential amino acids [Cat. no. 11140-036; Invitrogen], 1 mM sodium pyruvate [Cat. no. 11360-039; Invitrogen], 2 mM L-glutamine [Cat. no. 25030-024; Invitrogen], 1:100 penicillin–streptomycin [Cat. no. 15070; Invitrogen], and 0.1 mM 2-mercaptoethanol [Cat. no. 21985-023; Invitrogen]) with freshly added 50 ng/ml Bmp4 (Cat. no. 314-BP-010; R&D Systems), 50 ng/ml Bmp8a (Cat. no. 1073-BP-010; R&D Systems), 2 ng/ml SCF (Cat. no. 455-MC-010; R&D Systems), 500 ng/ml EGF (Cat. no. 2028-EG-010; R&D Systems), and 1,000 U/ml ESGRO (ESG1106; Millipore) to obtain a single cell suspension (1.5 × $10^5$ cells/ml). Cell suspension (100 $\mu$l/well) was added to 96 U-bottom well plates (Cat. no. 650970; Greiner-Bio) and incubated at 37°C, 5% $CO_2$ for 4 d.

### Colony forming assays

Clonal assays were performed as described previously (Chambers et al, 2003). Briefly, 600 cells in a single cell suspension were plated per 9.5 cm$^2$ and cultured for 6 d in serum medium and indicated concentrations of LIF (homemade). Formed colonies were fixed and stained for AP using the Leukocyte Alkaline Phosphatase Kit (86R-1KT; Sigma-Aldrich) according to the manufacturer's instructions.

## Molecular cloning of *Resf1*

To clone the *Resf1* coding sequence, total RNA extract was prepared from E14Tg2a ESCs using the RNeasy Mini Kit (74104; QIAGEN). The first cDNA strand was synthesized using oligo d(T) primers and Superscript III (18080093; Invitrogen). *Resf1* open reading frame was PCR amplified from the prepared cDNA in two overlapping parts. The primer overhangs introduced sites complementary to pBlue-Script plasmid into the 5′ and 3′ ends of the coding sequence. In addition, a triple flag tag and a glycine linker were inserted in front of the *Resf1* start codon. Both PCR products were subcloned into Blunt-TOPO vector using Zero Blunt TOPO PCR Cloning Kit (Cat. no. K2800-20SC; Invitrogen) and verified by Sanger sequencing. Two parts of the *Resf1* open reading frame were PCR amplified from the TOPO vectors and cloned into XhoI-, NotI-digested pPyPPGK plasmid (Chambers et al, 2003) using homemade Gibson assembly mix (50 mM Tris–HCl, pH 8.0, 5 mM $MgCl_2$, 0.1 mM dNTPs, 25 mU/$\mu$l Phusion polymerase, and 8 mU/$\mu$l T5 exonuclease).

## Episomal transfection

E14/T ESCs (Chambers et al, 2003) were transfected with pPyCAG-$Flag_3$-Resf1-PGK-Puro, pPyCAG-Flag3-Nanog-PGK-Puro, or pPyCAGPP (Chambers et al, 2003) plasmids using Lipofectamine 3000 (Cat. no. L3000001; Thermo Fisher Scientific). Transfected cells were cultured overnight in serum/LIF medium at 37°C, 7% $CO_2$. Cells were washed with PBS, dissociated using trypsin solution (as above), and analysed by a clonal assay (as above) in the presence of puromycin.

## Stable integration of transgenes into ESCs

To assess self-renewal of ESCs with ectopic expression of *Nanog*, *Esrrb*, or *DsRed* transgenes, linearised and purified pPyCAG-$(Flag)_3$-Nanog-IRES-Puro, pPyCAG-$(Flag)_3$-Esrrb-IRES-Puro and pPyCAG-DsRed-IRES-Puro plasmids (Chambers et al, 2003; Festuccia et al, 2012) were used to transfect *Resf1*$^{-/-}$ and E14Tg2a ESCs using Lipofectamine 3000 (Cat. no. L3000001; Invitrogen). Transfected cells were passaged every other day in serum/LIF medium supplemented with puromycin for six passages. Selected populations of cells were analysed by a colony forming assay in the presence of puromycin as described above.

## Deletion of *Resf1* gene in ESCs

To delete *Resf1*, two sets of four gRNAs (Table S1) targeting each end of the *Resf1* locus were cloned into eSpCas9(1.1)-T2A-eGFP and eSpCas9(1.1)-T2A-mCherry (#71814; Addgene) plasmids, respectively, as previously described (Ran et al, 2013). E14Tg2a ESCs were transfected with the eSpCas9 plasmids using lipofectamine 3000 (Cat. no. L3000001; Invitrogen). After 24 h, single cells expressing GFP and mCherry were isolated using fluorescence-activated cell sorting and expanded in serum/LIF medium. The isolated clonal cell lines were genotyped using primer pair A amplifying intergenic *Resf1* region and primer pair B which flanks the *Resf1* gene (Table S1). To verify *Resf1* deletion in individual alleles, PCR products were subcloned into Blunt-TOPO vector using Zero Blunt TOPO PCR Cloning Kit (Cat. no. K2800-20SC; Invitrogen) and sequenced using Sanger sequencing.

## Endogenous tagging of *Resf1*

To create endogenously tagged *Resf1* cell lines, a donor vector was constructed by cloning 1 kb 5′ and 3′ homology arms together with the $(v5)_3$-Stop-IRES-eGFP-STOP-loxP insert cassette into pBlue-Script plasmid. The homology arms were amplified from E14Tg2a genomic DNA by Q5 polymerase (Cat. no. M0491; NEB). The primers used introduced overhangs complementary to the insert cassette on the one side and pBlueScript on the other side. Homology arms and the insert cassette were cloned into EcoRI-HF (Cat. no. R3101S; NEB) cut pBlueScript using home-made Gibson assembly mix (described above). E14Tg2a ESCs were transfected with 1 $\mu$g of the donor vector and 1 $\mu$g of the eSpCas9(1.1)-T2A-Puro plasmid coding for a single gRNA targeting the *Resf1* stop codon (Table S1) using Lipofectamine 3000 (Cat. no. L3000001; Invitrogen). The next day, culture medium was replaced with serum/LIF medium supplemented with puromycin and cultured for one more day. Cells were washed twice with PBS and dissociated using trypsin solution (see above). Single cells with high GFP signal were isolated and expanded. DNA from the individual clonal cell lines was isolated using DNeasy Blood & Tissue Kits (Cat. no. 69504; QIAGEN) and genotyped using Rv5 genotyping primer pair (Table S1).

## RT-qPCR

ESCs were washed with PBS and lysed in 350 $\mu$l RLT buffer (QIAGEN) supplemented with 2-mercaptoethanol (1.4 M, Cat. no. M6250; Sigma-Aldrich). RNA was extracted using RNeasy Plus Mini kit (Cat. no. 74136; QIAGEN) according to the instructions. RNA was resuspended in nuclease-free $H_2O$ and stored at –80°C. Purified RNA was reverse transcribed using SuperScript III (Cat. no. 18080085; Invitrogen) according to the recommended protocol. Briefly, 0.2–1 $\mu$g of RNA was mixed with 50 ng of random hexamers (Cat. no. N8080127; Invitrogen) and 1 $\mu$l of 10 mM dNTPs (Cat. no. 10297018; Invitrogen) in a final volume of 11 $\mu$l. The mix was incubated at 65°C for 5 min followed by 2 min in ice. Next, 2 $\mu$l of 100 mM DTT, 20 U of RNAseOUT (Cat. no. 10777019; Invitrogen), and 100 U of SuperScript III reverse transcriptase (Cat. no. 18080044; Invitrogen) were added. The reaction mix was adjusted to the final volume of 20 $\mu$l with nuclease-free $H_2O$ and incubated 10 min at 25°C, 1 h at 50°C, and 15 min at 70°C. cDNA was diluted 1:10 (vol/vol) before quantification. The qPCR reaction mix was prepared by mixing 5 $\mu$l of the cDNA, 4.5 $\mu$l of Takyon SYBR 2X qPCR Mastermix (Eurogentec), and 0.5 $\mu$l of a primer pair mix (10 mM each) (Table S1) in a 384-well plate in duplicates. qPCR reaction was performed using 480 LightCycler (Roche). Specificity of the used primers was determined from a melting curve and their efficiency (>90%) by a linear regression.

## Co-immunoprecipitation

The E14/T ESCs (6 × $10^6$) were transfected with pPyCAG-$(HA)_3$-Nanog-IRES-Puro and pPyCAG-$(Flag)_3$-Resf1-IRES-Puro using Lipofectamine 3000 (Cat. no. L3000001; Invitrogen). In parallel, E14/T ESCs were transfected with only pPyCAG-$(HA)_3$-Nanog-IRES-Puro plasmid as a negative control. The day after the transfection, medium was replaced with fresh serum/LIF medium supplemented with G418 and puromycin and cultured for 1 d. Cells were detached

by trypsin, washed with PBS, and collected. Cells were burst in a hypotonic buffer (5 mM Pipes, pH 8, and 85 mM KCl) with freshly added 0.5% NP-40 and 1× Protease Inhibitor Cocktail (Cat. no. 4693159001; Roche) for 20 min in ice. Nuclei were collected by centrifugation (830 $g$, 5 min, 4°C), resuspended in 1 ml of the NE buffer (20 mM Hepes, pH 7.6, 350 mM KCl, 0.2 mM EDTA, pH 8, 1.5 mM MgCl$_2$, and 20% glycerol) with freshly added 0.2% NP-40, 0.5 mM DTT, and 1× Protease Inhibitor Cocktail (Cat. no. 4693159001; Roche) and transferred into NoStick microtubes (Cat. no. LW2410AS; Alpha Laboratories). Nuclei were lysed for 1 h at 4°C in the presence of 150 U of benzonase nuclease (Cat. no. 71206; Novagen) while rotating. Nuclear lysates were cleared by centrifugation (17,000 $g$, 30 min, 4°C) and transferred into clean NoStick tubes. Input control fractions (5%) were separated. Anti-FLAG M2 affinity beads slurry (30 $\mu$l, Cat. no. A2220; Sigma-Aldrich) was washed three times with PBS and resuspended in the original volume of the NE buffer. Thirty microlitres of the beads were added to the remaining nuclear extracts and incubated for 2 h at room temperature on a wheel. The beads were washed three times with PBS using magnet. Proteins were eluted by boiling the beads for 5 min in 30 $\mu$l of 1× LDS Sample Buffer (Cat. no. B0008; Invitrogen) supplemented with 250 mM DTT. The elution was repeated, and the two fractions were combined. The input and immunoprecipitated samples were analysed by SDS–PAGE and immunoblot.

### SDS–PAGE and immunoblotting

The protein extracts were mixed with 4X Bolt LDS Sample Buffer (Cat. no. B0008; Invitrogen) and 250 mM DTT and boiled for 5 min. Samples together with SeeBlue Plus2 pre-stained protein standard (Cat. no. LC5925; Invitrogen) were loaded on a Bolt 12% Bis-Tris Plus Gel (Cat. no. NW00122BOX; Life Biosciences) and run at constant 180 V in a Bolt MES SDS running buffer (Cat. no. B0002; Invitrogen). The proteins were transferred onto a nitrocellulose membrane in transfer buffer (25 mM Tris–HCl, pH 8, 0.21 M glycine, and 10% methanol) at constant 180 mA overnight at 4°C or at 380 mA for 70 min in ice. The membrane was blocked in 10% skimmed milk resuspended in PBS and supplemented with 0.1% Tween-20 (PBS-T) for 1 h or overnight. Primary antibodies (Table S1) were diluted in 5 ml of 5% milk in PBS-T and added on the membrane. The membrane was stained for at least 1 h at the room temperature while swirling. The membrane was washed four times for 5 min with PBS-T. The secondary antibodies (Table S1) were diluted in 5 ml of 5% milk in PBS-T and incubated with the membrane for at least 1 h at the room temperature. The membrane was washed four times for 5 min in PBS-T. The membrane was incubated with Pierce ECL Western Blotting Substrate (Cat. no. 32106; Invitrogen) for 2 min at room temperature if an antibody with conjugated horse radish peroxidase was used.

### Immunostaining

Cells were washed with PBS and fixed with 4% paraformaldehyde at room temperature for 10 min. Cells were washed with PBS and permeabilised in 0.3% vol/vol Triton X-100 in PBS for 15 min at room temperature. The solution was discarded, and cells were blocked in 0.1% Triton X-100 and 3% vol/vol donkey serum in PBS for 1 h at

room temperature. Fixed and permeabilised cells were incubated with primary antibodies (Table S1) diluted in blocking buffer overnight at 4°C. Cells were washed four times with PBS containing 0.1% Triton X-100 and incubated with the fluorophore-labelled secondary antibodies diluted (1:1,000 vol/vol) in blocking buffer for 1 h at room temperature. DNA was stained with 4′,6-diamidino-2-phenylindole (DAPI; 1:2,000 dilution in PBS) for 5–10 min at room temperature. DAPI was washed once with PBS for 5 min and samples were stored in a mounting solution at 4°C in dark. Samples were analysed using SP8 Lightning confocal microscope (Leica).

### Flow-cytometry

To quantify SSEA1 and CD61 expression, cell aggregates were collected, washed with PBS and dissociated in 0.1% trypsin solution at 37°C for 10 min. Trypsin was neutralised with serum medium and passed through a cell strainer. A small proportion of each analysed suspension was combined in a separate tube for control samples. Cells were centrifuged (300 $g$, 3 min) and resuspended in 100 $\mu$l of serum medium supplemented with 0.5 $\mu$l SSEA-I (Cat. no. 125607; BioLegend) and 0.15 $\mu$l PE-CD61 antibodies (Cat. no. 104307; BioLegend). The negative control was resuspended in 300 $\mu$l of the serum medium and divided into three 100 $\mu$l fractions. SSEA-I and CD61 antibodies were added to one fraction resulting in two single-stain control samples and one negative control. Cells were incubated for 15 min at room temperature in dark and washed twice with PBS. Cells were resuspended in 200 $\mu$l of 2% KSR in PBS and analysed on a 5 laser LSR Fortessa analyser (BD Biosciences). Single cells were gated based on forward and side scatters. Live cells were gated based on DAPI signal and the SSEA-I, CD61 double-positive population was gated based on the negative and single stain controls.

### Analysis of RNA sequencing data

RNA sequencing datasets from wild-type and $Resf1^{-/-}$ ESCs (Fukuda et al, 2018) were downloaded from sequence read archive (SRR6423833, SRR6423836, SRR6423832, SRR6423839) and pseudo-counts for each transcript were quantified using Salmon v1.5.2 (Patro et al, 2017) using default settings and index generated from GRCm38 mouse transcriptome. The differential expression analysis was performed using tximport v1.18.0 (Soneson et al, 2016) and DEseq2 v1.30.0 (Love et al, 2014) R packages for each transcript. Estimated $\log_2$ fold-change between the mean expression of ENSMUST00000171588 transcript in wild-type and $Resf1^{-/-}$ and corresponding adjusted $P$-value were reported in the text. To visualise relative expression of ENSMUST00000171588 LIFR transcript in wild-type and $Resf1^{-/-}$ ESCs, transcripts with less than one transcript per million in more than one sample were removed and the pseudo counts were normalised using trimmed mean of M-values (TMM) and counts were normalised per million of sequenced reads using edgeR (Robinson et al, 2010). To generate the volcano plot, per-gene pseudo counts were loaded using tximport v1.18.0 (Soneson et al, 2016) and genes with less than 10 counts in all samples combined were removed. Differential analysis per gene was performed by DEseq2 v1.30.0 (Love et al, 2014) and the fold-changes were shrank using apeglm method (Zhu et al, 2019). Genes

with absolute corrected fold-change higher than 1.5 and adjusted *P*-value lower than 0.05 were considered as differentially expressed. Plots were made using ggplot2 v3.3.3 (Wickham et al, 2019).

### Analysis of single cell RNA sequencing data

Single cell RNA sequencing data of mouse embryo were obtained using R package MouseGastrulationData (Griffiths & Lun, 2020). The number of individual cells expressing *Resf1* (log normalised counts > 0) was determined in epiblast and PGCs at indicated embryonic stages. *Resf1* expression in epiblast and PGCs at different embryonic stages was visualised by plotting log normalised counts of *Resf1* in cells with *Resf1* log normalised counts > 0.

### Statistical analysis

Methods used for statistical tests and summary statistics used for visualisation are indicated for each figure. The Benjamini–Hochberg method was used to correct for multiple testing. All statistical analyses were performed using R programming language (v4.0.3) and Rstatix package (v0.7.0).

## Data Availability

No data were deposited in a public database.

## Supplementary Information

## Acknowledgements

We thank members of the Chambers lab for discussions. We are grateful to Elisa Barbieri and Dougie Colby for technical assistance. This work was funded by a Medical Research Council (UK) grant to I Chambers (MR/T003162/1) and M Vojtek was supported by a Principal's Career Development Scholarship from the University of Edinburgh.

### Author Contributions

M Vojtek: conceptualization, data curation, software, formal analysis, validation, investigation, visualization, methodology, project administration, and writing—original draft, review, and editing.
I Chambers: conceptualization, resources, supervision, funding acquisition, and writing—original draft, review, and editing.

### Conflict of Interest Statement

The authors declare that they have no conflict of interest.

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
