## [Reviewer comments · Life Science Alliance]

Life Science Alliance

Loss of Resf1 reduces the efficiency of embryonic stem cell self-renewal and germline entry

Matúš Vojtek and Ian Chambers

DOI: <https://doi.org/10.26508/lsa.202101190>

Corresponding author(s): Ian Chambers, University of Edinburgh

Review Timeline:

Submission Date:	2021-08-12
Editorial Decision:	2021-08-13
Revision Received:	2021-09-14
Editorial Decision:	2021-09-21
Revision Received:	2021-09-23
Accepted:	2021-09-23

Scientific Editor: Novella Guidi

Transaction Report:

Please note that the manuscript was reviewed at *Review Commons* and these reports were taken into account in the decision-making process at *Life Science Alliance*.

Review
COMMONS

August 13, 2021

Re: Life Science Alliance manuscript #LSA-2021-01190

Prof. Ian Chambers
Edinburgh, University of
MRC Centre for Regenerative Medicine
5 Little France Dr
University of Edinburgh
Edinburgh, SCOTLAND EH16 4UU
United Kingdom

Dear Dr. Chambers,

Thank you for submitting your manuscript entitled "Resf1 deletion reduces the efficiency of embryonic stem cell self-renewal and germline entry" to Life Science Alliance. The manuscript was submitted and reviewed via Review Commons. The authors then chose to transfer their somewhat revised manuscript, along with the reviewers' comments and a proposed revised plan to Life Science Alliance (LSA). The reviewer comments and revision plan was assessed at LSA, and LSA editors deemed that the manuscript could be further considered at LSA provided the authors revise the manuscript, in accordance to what they have laid out in the pbp rebuttal / revision plan. We, thus, encourage you to submit a revised manuscript to us that includes all the changes you have laid out in your Revision plan. Given that new data will be added to the revised manuscript, the revision will have to be looked at by a set of referees, most likely the same ones as Review Commons.

Thank you for this interesting contribution to Life Science Alliance. We are looking forward to receiving your revised manuscript.

Sincerely,

- A letter addressing the reviewers' comments point by point.
- An editable version of the final text (.DOC or .DOCX) is needed for copyediting (no PDFs).
- High-resolution figure, supplementary figure and video files uploaded as individual files: See our detailed guidelines for preparing your production-ready images, <https://www.life-science-alliance.org/authors>
- Summary blurb (enter in submission system): A short text summarizing in a single sentence the study (max. 200 characters including spaces). This text is used in conjunction with the titles of papers, hence should be informative and complementary to the title and running title. It should describe the context and significance of the findings for a general readership; it should be written in the present tense and refer to the work in the third person. Author names should not be mentioned.

B. MANUSCRIPT ORGANIZATION AND FORMATTING:

Response to Reviewer's Comments

Reviewer #1 (Evidence, reproducibility and clarity (Required)):

This paper puts together a nice set of data showing that a specific gene called *Resf1* when deleted affects the ability of ESCs to self-renew and proceed to germline fates. I believe the data are sound and that they provide the evidence needed for the authors to make their conclusions.

Reviewer #1 (Significance (Required)):

While I think the data are presented well and the manuscript is well-written, the "modest" functional results suggest this work would be more suited for a specialized journal.

We thank Reviewer 1 for their supportive comments.

Reviewer #2 (Evidence, reproducibility and clarity (Required)):

The authors uncovered the new roles of *Resf1* in mESC self-renewal and germline entry. They showed that *Resf1* deletion reduced mESC self-renewal, and it's not required for *Nanog* function. In addition, the efficiency of PGCLC specification of *Resf1* knockout mESC is less than WT mESC. However, these conclusions are too preliminary and the underlying mechanism is missing.

****Major comments:****

1. In the presence of LIF, there is no difference between *Resf1* knockout mESCs and WT mESCs except the expression of *Esrrb*, *Nanog* and *Pou5f1*. What about other genes? RNA-seq is needed to distinguish the two cell lines.

Fukuda et al. have shown by RNA-seq that deletion of Resf1 leads to misregulation of ~1000 genes (adj. p-value < 0.05, fold change > 2). This highlights large differences between transcriptomes of Resf1 KO and WT cells that occur despite only a marginal difference in self-renewal efficiency between Resf1 KO cells and WT in the presence of LIF.

*We have reanalysed the Fukuda et al data. In agreement with our Q-RT-PCRs this suggests that *Esrrb* is significantly downregulated, as are *Klf4* and *LifR* (FDR <0.05, fold change > 1.5). However, *Pou5f1* and *Nanog* were not differentially expressed (FDR <0.05, fold change > 1.5). This is in line with the lower level of downregulation of *Pou5f1* and *Nanog*, compared to *Esrrb* in our Q-RT-PCR data. Notably, our gene expression analyses were performed in 5 biological replicates, whereas Fukuda et al. performed RNA-seq in two biological replicates. We have included our analysis of the Fukuda et al data in our revised submission (Figures 2D and 3B). Given the fact we see a change in ESC self-renewal at low LIF concentrations, we have carefully analysed expression of *LifR* in the RNA-seq data. This suggests that the major change in expression of *LIFR* mRNA isoforms is a decrease in expression of transmembrane *LIFR* mRNA. We have directly assessed expression of *LIFR* mRNAs using a Q-RT-PCR strategy that discriminates between expression of the transmembrane *LifR* and soluble *LifR* (Chambers, BJ, 1997). This is important since soluble *LIFR* acts antagonistically. These data are now presented as Figure 2 panels D and E. Together they show a similar 2.5-fold decrease in expression of both transmembrane and soluble forms of *LifR* mRNAs. This co-ordinate decrease is consistent with a mechanism in which *RESF1* acts (directly or indirectly) to stimulate transcription of the *LIFR* gene.*

2. The authors showed *Resf1* is not required for *Nanog* function, so how does *Resf1* regulate the

expression of pluripotency genes? Through epigenetic modifications or signaling pathways? The authors should design experiments to explain the detailed mechanisms.

This are important questions to answer. However, many more experiments would be required to reach firm conclusions. The reviewer is right to say that the mechanisms by which Resf1 affects pluripotency are unknown and remain to be answered in future. We have therefore updated the text to discuss similarities in pluripotency phenotype between deletions of Trim28, SETDB1, YTHDC1 and RESF1. As deletion of the RESF1 partner SETDB1 or other proteins involved in repression of retrotransposons lead to downregulation of pluripotency genes and in some cases collapse of ESCs (e.g PMID: 19884255, Bilodeau et al. 2009; PMID: 19884257, Yuan et al. 2009), we hypothesise that the RESF1 phenotype may be explained by affecting SETDB1 chromatin binding and therefore repression of SETDB1 targets. The mild phenotype of RESF1 KO indicates that RESF1 is not an essential component of this repressor complex.

3. The authors showed that Resf1 interacts with Nanog, but they used forced expressed proteins. Does the endogenous Resf1 interacts with endogenous Nanog? Do they bind to some same DNA sequences?

The strength of the immunoblot signal for RESF1 is low, even when Resf1 is expressed episomally. Therefore, although we could try to co-immunoprecipitate with the Resf1-v5 cell line and endogenous Nanog, this seems like an unnecessary amount of effort given the fact that the result will not affect the conclusions of our study.

4. In figure 5C, some Resf1 positive cells showed Nanog negative. Are these Nanog negative cells pluripotent?

Nanog-null ESCs are pluripotent (PMID: 18097409, Chambers et al., 2007). In addition NANOG-negative cells in FCS/LIF cultures can retain pluripotency. Our purpose in this figure was therefore not to say whether NANOG-negative:RESF1-positive cells are pluripotent but to draw attention to the broader expression of RESF1 in FCS/LIF compared to NANOG. Such broader expression has also been noted for other heterogeneously expressed factors (PMID: 31582397, Pantier et al. 2019).

We have reworded the writing in the text accordingly.

5. In figure 6A, the naïve mESCs are induced to EpiLCs. Is the transition efficiency of Resf1 knockout cells the same with WT mESCs? The finally obtained PGCLCs should be identified.

We show that the key TFs of EpiLC state are expressed similarly in WT and Resf1 KO cells (Supplementary figure 4). We have added photographs of EpiLC colonies formed by Resf1 KO and WT cells that shows that WT and Resf1KO EpiLCs have a similar morphology. Together this suggests an efficient transition to an EpiLC state.

Our analysis has identified expression of Blimp1/Ap2g/Prdm14 in Resf1-null cultures. Compared to wild-type cells these levels are reduced up to 3-fold. As this is from an unsorted population and the number of SSEA1/CD61-positive cells is decreased around 2x, this suggests that the PGCLC population formed by Resf1-null cells is reduced in proportion but is otherwise normal.

****Minor comments:****

1. in figure 5c, the scale bar is missing.
The missing scalebar has been added.

Reviewer #2 (Significance (Required)):

The authors uncovered the new roles of Resf1 in mESC self-renewal and germline entry.

Reviewer #3 (Evidence, reproducibility and clarity (Required)):

The authors aimed to study RESF1 in ESC's to understand it's role in germ cell specification and PGCLC differentiation under in vitro experimental challenges.

The experiments performed and reported were thorough and convincing. Data and methods were clearly explained.

What was less clear was an explanation of why colonies 4 and 24 were chosen. Were there other colonies with the desired expression? Was this amount of expression repeated in replicative experiments with approximately 2 colonies only available to be selected?

30 colonies were selected for analysis. Of these, only 2 had deletion of both Resf1 alleles. This point has been made clearer in the text.

Figure 1C, 5C ad S2B with microscopic images should include a scale bar.

Unfortunately the microscopy setup used to collect the images in Figures 1C (now 2A) and S2B did not allow scale bars to be added at the time of imaging and these cannot be added retrospectively. However, we do not think that inclusion of scale bars, even were it possible would affect the conclusions of our manuscript. Missing scale bars in the Figure 5C have been added.

Figure 1E needs a better explanation of the significance, "less clear cut" is not adequate. Reporting statistics, or lack of significance, on the graph would help.

We have updated the manuscript and the Figure 2C to include results of a statistical analysis (Wilcoxon-rank sum test) comparing formation of AP+ colonies between Resf1 KO and WT cells at different LIF concentrations. These results show that Resf1 KO cells have lower median number of AP+ colonies than WT cells at LIF concentrations 0, 1 and 3 U/ml (p.adj. < 0.05). The differences between RESF1 KO and WT cells were not significant at higher concentrations of LIF.

Reviewer #3 (Significance (Required)):

Understanding the specific interactions and suggested role of RESF1 in self-renewal is informative on a molecular biology and developmental biology level.

It's translatability to medicine, although perhaps that is not the intention, is somewhat lacking. Is there a naturally occurring situation where LIF is absent that would require this pathway to be used?

These were mouse ESC's, perhaps this study could incorporate information about relevant translation to a human condition to aid in the significance. This manuscript suggests a mechanistic evaluation by which self-renewal can occur other than the canonical pathway, which is interesting and can inform the field.

Our results suggest that RESF1 directly or indirectly supports self-renewal of ESCs. Interestingly, Human cell atlas identified RESF1 expression as a negative predictor of survival of renal cancer and was found to be expressed in testis cancer cells and other cancer tissues. Therefore, RESF1 could promote self-renewal of cancer cells similarly to ESCs. However, this is speculative and needs further studies. As this is both outside of the scope of this manuscript and our expertise, we do not think it prudent for us to pursue this line of inquiry. However, we agree that further studies could evaluate RESF1 function in human tissues, especially pluripotent cells and germ cells. As we show that RESF1 deletion leads to reduced induction of PGCLCs and previous studies showed infertility of Resf1 KO mice, investigating link between human fertility and RESF1 could have implications in reproductive medicine.

We have updated our discussion to highlight the possible significance of RESF1 function in human fertility.

September 21, 2021

RE: Life Science Alliance Manuscript #LSA-2021-01190R

Prof. Ian Chambers
University of Edinburgh
Centre for Regenerative Medicine, Institute for Regeneration and Repair
5 Little France Drive
University of Edinburgh
Edinburgh, Scotland EH16 4UU
United Kingdom

Dear Dr. Chambers,

Thank you for submitting your revised manuscript entitled "Loss of Resf1 reduces the efficiency of embryonic stem cell self-renewal and germline entry". We would be happy to publish your paper in Life Science Alliance pending final revisions necessary to meet our formatting guidelines.

- please add the Twitter handle of your host institute/organization as well as your own or/and one of the authors in our system
- please consult our manuscript preparation guidelines <https://www.life-science-alliance.org/manuscript-prep> and make sure your manuscript sections are in the correct order and properly labeled
- please separate the Figure legends and Supplemental Figure legends into separate sections;
- please add a callout for Figure S5 to your main manuscript text
- please add size markers next to the blots in figure 5
- please add scale bars to Figure 2A, and indicate their size in the legend

A. FINAL FILES:

B. MANUSCRIPT ORGANIZATION AND FORMATTING:

Sincerely,

Reviewer #1 (Comments to the Authors (Required)):

The authors have answered my questions.

September 23, 2021

RE: Life Science Alliance Manuscript #LSA-2021-01190RR

Prof. Ian Chambers
University of Edinburgh
Centre for Regenerative Medicine, Institute for Regeneration and Repair
5 Little France Drive
University of Edinburgh
Edinburgh, Scotland EH16 4UU
United Kingdom

Dear Dr. Chambers,

Thank you for submitting your Research Article entitled "Loss of Resf1 reduces the efficiency of embryonic stem cell self-renewal and germline entry". It is a pleasure to let you know that your manuscript is now accepted for publication in Life Science Alliance. Congratulations on this interesting work.

DISTRIBUTION OF MATERIALS:

Again, congratulations on a very nice paper. I hope you found the review process to be constructive and are pleased with how the manuscript was handled editorially. We look forward to future exciting

submissions from your lab.

Sincerely,
